# Renal and major clinical outcomes and their determinants after nephrectomy in patients with pre-existing chronic kidney disease: A retrospective cohort study

Maxime Schleef[1,2]*, Pascal Roy[3], Sandrine Lemoine[1,4], Philippe Paparel[5], Marc Colombel[6], Lionel Badet[6], Fitsum Guebre-Egziabher[1,7]

1 Lyon University, CarMeN laboratory, IRIS team, INSERM, INRAE, Université Claude Bernard Lyon-1, Bron, France, 2 Department of intensive care medicine, Hôpital Edouard Herriot, Hospices Civils de Lyon, Lyon, France, 3 Department of biostatistics-bioinformatics, Pôle Santé Publique, Hospices Civils de Lyon, Lyon, France, 4 Department of renal explorations, Hôpital Edouard Herriot, Hospices Civils de Lyon, Lyon, France, 5 Department of urology, Centre Hospitalier Lyon Sud, Hospices Civils de Lyon, Pierre-Bénite, France, 6 Department of urology and transplantation surgery, Hôpital Edouard Herriot, Hospices Civils de Lyon, Lyon, France, 7 Department of nephrology-hypertension-dialysis, Hôpital Edouard Herriot, Hospices Civils de Lyon, Lyon, France

* maxime.schleef@chu-lyon.fr

**Data Availability Statement:** All relevant data are within the paper and its Supporting Information files.

## Abstract

The consequences of partial nephrectomy (PN) compared to radical nephrectomy (RN) are less documented in patients with pre-existing chronic kidney disease (CKD) or with solitary kidney (SK). We assessed renal outcomes, and their determinants, after PN or RN in a retrospective cohort of patients with moderate-to-severe CKD (RN-CKD and PN-CKD) or SK (PN-SK). All surgical procedures conducted between 2013 and 2018 in our institution in patients with pre-operative estimated glomerular filtration rate (eGFR)<60 mL/min/1.73m² or with SK were included. The primary outcome was a composite criterion including CKD progression or major adverse cardio-vascular events (MACE) or death, assessed one year after surgery. Predictors of the primary outcome were determined using multivariate analyses. A total of 173 procedures were included (67 RN, and 106 PN including 27 SK patients). Patients undergoing RN were older, with larger tumors. Preoperative eGFR was not significantly different between the groups. One year after surgery, PN-CKD was associated with lower rate of the primary outcome compared to RN-CKD (43% vs 71% p = 0.007). In multivariate analysis, independent risk factors for the primary outcome were postoperative AKI (stage 1 to stage 3 ranging from OR = 8.68, 95% CI 3.23–23.33, to OR = 28.87, 95% CI 4.77–167.61), larger tumor size (OR = 1.21 per cm, 95% CI 1.02–1.45), while preoperative eGFR, age, sex, diabetes mellitus, and hypertension were not. Postoperative AKI after PN or RN was the major independent determinant of worse outcomes (CKD progression, MACE, or death) one year after surgery.

**Funding:** MS was partially supported by a grant from the Hospices Civils de Lyon [Année Médaille d'Or 2021] [https://www.chu-lyon.fr]. The funder had no role in study design, data collection and analysis, decision to publish, or preparation of the manuscript.

**Competing interests:** The authors have declared that no competing interests exist.

## Introduction

Nephron sparing strategies (NSS), such as partial nephrectomy (PN), have progressively replaced historical radical nephrectomy (RN) for treatment of kidney tumor, becoming the standard procedure for small renal masses, and whenever technically feasible for larger ones [1–3]. Indeed, retrospective studies have shown that PN, compared to RN, improved overall and cancer-related mortality, and was associated with fewer cardio-vascular events [4–8]. However, the only randomized controlled trial (EORTC trial) comparing PN and RN reported that PN resulted in better renal function [9] but did not improve oncological outcomes or overall survival [9, 10]. It is therefore considered that PN allows to preserve nephron capital, with at least equivalent oncological outcomes and survival. In parallel, mini-invasive approaches, including laparoscopic and more recently robot-assisted laparoscopic procedures, have also been increasingly used. Retrospective studies have shown that mini-invasive NSS resulted in non-inferior oncological outcomes compared to open NSS, with fewer post-operative complications [11], with some studies suggesting a benefit of robot-assisted over laparoscopic-only procedures [12–14]. NSS and mini-invasive techniques have consequently been used in patients with chronic kidney disease (CKD) or with solitary kidney (SK) in order to preserve renal parenchyma. However, the need of temporary vascular clamping during PN has raised concern about the risk of renal ischemia-reperfusion injuries, potentially leading to acute kidney injury (AKI) and worsening of CKD. Retrospective studies have reported an incidence of post-operative AKI following PN with renal clamping that ranged from 20 to 40%, associated with a worse renal function 1 year after surgery [15–17]. Robot-assisted PN seemed interesting since it was associated in some studies with a shorter ischemia time than open or laparoscopic PN [18, 19] but studies failed to demonstrate its benefit on renal function preservation [11, 12, 18, 20]. In a recent prospective randomized study, super selective clamping of tumor-targeted arteries also failed to provide better renal function preservation compared to conventional robot-assisted PN [21], in addition, achieving zero ischemia during NSS through "off clamp" procedures has also been proposed to avoid renal ischemia reperfusion injuries, but it remains technically demanding [22].

Little is known about the effects on renal function of these recent surgical procedures in the specific population of patients with CKD. Retrospective studies have reported a beneficial effect of PN on CKD progression in stage 3A CKD patients compared to RN [23], or no effect [6], and on the requirement of permanent dialysis in severe CKD patients [24]. PN was also associated with lower renal function deterioration in CKD patients compared to those with normal kidney function [25]. These studies were limited by short [25] or incomplete [6] follow-up, lack of power [24], or the exclusion of severe CKD [6, 23].

We therefore conducted a retrospective study to assess the renal and clinical outcomes after a RN or PN in patients with pre-existing moderate-to-severe CKD or with SK. The second aim was to identify the major determinants of adverse outcomes in this population.

## Methods

### Patients

The retrospective Outcomes after Nephrectomy in patients with CKD (ON-CKD) cohort study reported herein included adult patients hospitalized in the Hospices Civils de Lyon, France. Inclusion criteria were any surgical procedure (RN or PN) for a renal tumor from January 2013 to December 2018, with a baseline estimated glomerular filtration rate (eGFR) < 60 mL/min/1.73m$^2$ prior to surgery (RN-CKD and PN-CKD groups), or a PN for a renal tumor on a SK regardless of eGFR (PN-SK group). The criteria for choosing PN or RN were based on

french guidelines at the time, in which PN was the first option whenever feasible [26]. Exclusion criteria were a baseline eGFR < 15 mL/min/1.73m$^2$ or maintenance hemodialysis or peritoneal dialysis, kidney transplantation, ablative therapy (cryo- or thermo-ablation), and RN performed in SK.

## Baseline characteristics

Data were collected from electronic medical records, that were accessed for research purpose from January to June, 2021, with access to information that could potentially identify individual participants during data collection but then anonymously analyzed. At baseline, demographic characteristics, previous medical history, known follow-up by a nephrologist, and the etiology of CKD, were collected. The most recent eGFR according to serum creatinine in the 6 months prior to surgery was considered baseline (using the Chronic Kidney Disease Epidemiology Collaboration [CKD-EPI] equation, or the Modification of Diet in Renal Disease [MDRD] equation only when the former was not applicable, i.e. with older creatinine measured without IDMS traceable method).

## Tumor, surgery, and hospitalization

Characteristics of the tumor (histopathological report), characteristics of the surgical procedure, and post-operative events during hospitalization were collected. Intra- and post-operative complications were graded according to the Clavien-Dindo classification [27] and only severe complications (Clavien-Dindo >2) were considered for the present study.

Post-operative AKI during the first post-operative week was defined according to the creatinine elevation criterion of KDIGO 2012 classification (urine output criterion was not used as it was not recorded) [28].

## Outcomes

Clinical outcomes and events were collected one month and one year after surgery, in electronic medical records. The primary outcome was a composite endpoint at one year including CKD progression, major adverse cardio-vascular events (MACE), and all-cause mortality. CKD progression was defined as an increase of at least 1 CKD stage, or initiation of renal replacement therapy (maintenance hemodialysis or peritoneal dialysis), using the closest known eGFR to the 1-year post-operative date. MACE included myocardial infarction, unstable angina, cardiac decompensation, or stroke, and as reported in the medical record. Secondary outcomes included all-cause death, cancer-related death, MACE, renal events (CKD progression, decline of eGFR from baseline, preservation of eGFR >90% of baseline value) at one month and one year after surgery. Missing data were removed from the corresponding analyses.

## Statistics

Categorical and continuous variables were, respectively, expressed as numbers and percentages or median with interquartile ranges [IQR], and compared with the Chi-squared or Mann-Whitney test. Univariate analysis of the primary outcome used the Chi-squared test. The corresponding multivariate analysis was performed fitting unconditional logistic regression model. Nested models were compared using likelihood ratio tests. A similar approach was applied for the analyses to the binary secondary renal outcomes. The RN-CKD and PN-CKD eGFR mean variations were compared using the Student T-test. The multivariate analysis of the eGFR determinants (for the total population of all groups) at one year post surgery was

performed fitting a multivariate linear regression. Nested models were compared using ANOVA. Covariates of interest and previously identified as risk factors in the literature (age, sex, diabetes mellitus, hypertension, preoperative eGFR, postoperative AKI and tumor diameter) [15–17, 29] were assessed for collinearity and interaction, included in the multivariate models and then tested in a backward-stepwise process. In all statistical tests (two-tailed), p-values smaller than 5% were considered as significant. Statistical analyses were conducted using SPSS Statistics (IBM, Armonk, NY, US) or GraphPad Prism v6.0 (San Diego, CA, US).

### Ethics

The ON-CKD study was conducted in accordance with the Declaration of Helsinki, and it was approved by the ethics committee/institutional review board of *Comité Scientifique et Ethique des Hospices Civils de Lyon* (*IRB number* 00013204, *numéro avis* 20_050, *numéro registre* CNIL 19_388). In accordance with French Law about retrospective studies on observational data, written consent was waived by the ethics committee/institutional review board of *Comité Scientifique et Ethique des Hospices Civils de Lyon* (*IRB number* 00013204, *numéro avis* 20_050, *numéro registre* CNIL 19_388) but patients were informed of the study and had the right to oppose the use of their data.

## Results

### Patient characteristics

Among 1259 surgical procedures conducted for a renal mass, 173 (171 patients) were included: there were a total of 67 RN in CKD patients (RN-CKD) and 106 PN; among the latter 79 were in CKD patients (PN-CKD) and 27 were in SK (PN-SK; Fig 1). Patients undergoing RN-CKD were older (median age 74 [68–79] vs 68 [64–76] years, $p = 0.004$) and had lower rate of hypertension (60 vs 80%, $p = 0.02$) compared to PN-CKD (Table 1).

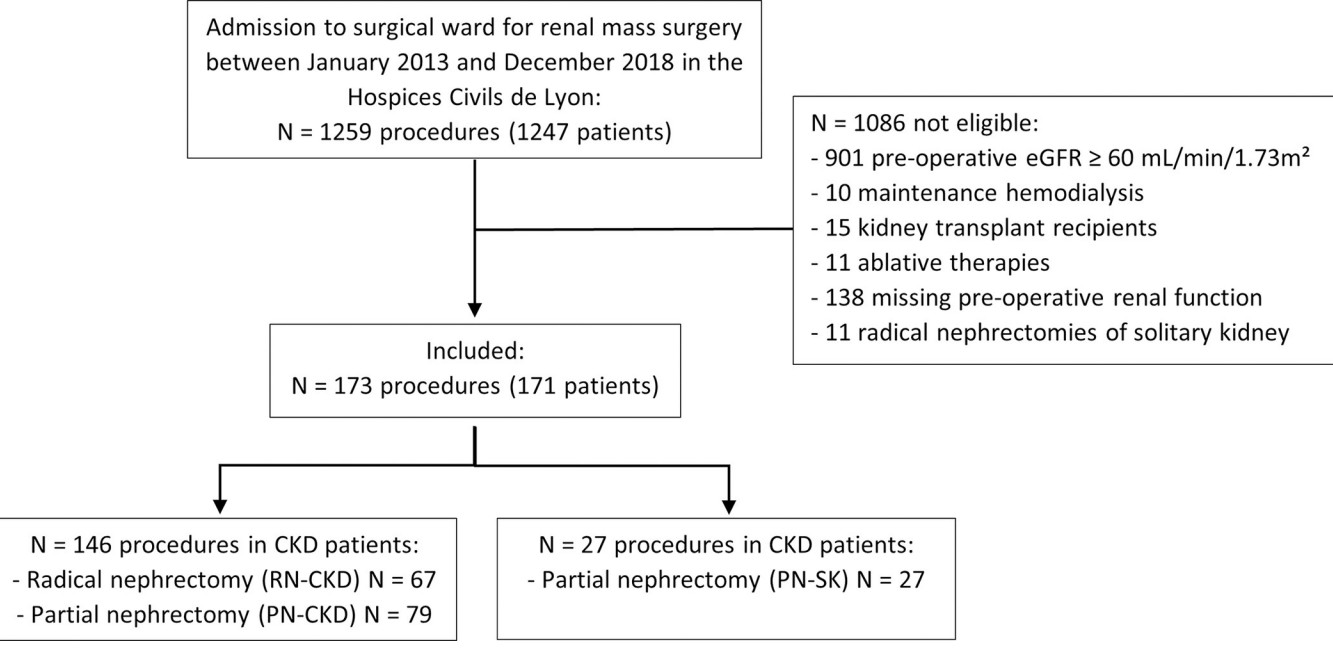

**Fig 1. Study flow-chart.** *eGFR* estimated glomerular filtration rate, *CKD* chronic kidney disease.

**Table 1. Baseline patient characteristics.**

| | CKD patients (*n* = 146) | | *p*-value | SK patients | *p*-value | |
|---|---|---|---|---|---|---|
| | RN-CKD (*n* = 67) | PN-CKD (*n* = 79) | | PN-SK (*n* = 27) | vs. RN-CKD | vs. PN-CKD |
| Age years | 74 [68–79] | 68 [64–76] | 0.004 | 68 [58–77] | 0.006 | NS |
| Sex male (%) | 44 (66) | 55 (70) | NS | 15 (56) | NS | NS |
| BMI kg/m² | 25.6 [23.9–29] | 27.5 [24.3–31.1] | NS | 27.6 [23.4–31.1] | NS | NS |
| ASA score (%): | | | | | | |
| • ASA 1 | 5 (7) | 4 (5) | NS | 3 (11) | NS | NS |
| • ASA 2 | 32 (48) | 47 (59) | NS | 13 (48) | NS | NS |
| • ASA 3 | 26 (39) | 17 (22) | NS | 8 (30) | NS | NS |
| • Unknown | 4 (6) | 11 (14) | NS | 3 (11) | NS | NS |
| Previous medical history (%): | | | | | | |
| • Active smoker | 6 (9) | 10 (13) | NS | 4 (15) | NS | NS |
| • Alcoholism | 1 (1) | 1 (1) | NS | 0 | NS | NS |
| • Diabetes mellitus | 19 (28) | 30 (38) | NS | 4 (15) | NS | NS |
| • Hypertension | 40 (60) | 63 (80) | 0.008 | 21 (78) | NS | NS |
| • Cardiac failure | 4 (6) | 8 (10) | NS | 0 | NS | NS |
| • Stroke | 3 (4) | 3 (4) | NS | 1 (4) | NS | NS |
| • Coronary artery disease | 8 (12) | 9 (11) | NS | 1 (4) | NS | NS |
| • Peripheral artery disease | 5 (7) | 2 (3) | NS | 0 | NS | NS |
| • Sleep apnea | 7 (10) | 7 (9) | NS | 4 (15) | NS | NS |
| • Respiratory failure | 1 (1) | 1 (1) | NS | 0 | NS | NS |
| Baseline SCr µmol/L | 117 [102–140] | 123 [111–136] | NS | 104 [89–132] | NS | NS |
| eGFR mL/min/1.73m² | 50 [41–55] | 49 [43–55] | NS | 54 [47–68] | NS | NS |
| CKD stage (%): | | | | | | |
| • 1–2 | 0 | 0 | NS | 11 (40) | <0.001 | <0.001 |
| • 3A | 44 (66) | 57 (72) | NS | 10 (37) | 0.01 | 0.001 |
| • 3B | 16 (24) | 17 (22) | NS | 4 (15) | NS | NS |
| • 4 | 7 (10) | 5 (6) | NS | 2 (7) | NS | NS |
| Cause of CKD (%): | | | | | | |
| • Diabetes | 2 (3) | 4 (5) | NS | 2 (7) | NS | NS |
| • Hypertension | 6 (9) | 17 (22) | NS | 2 (7) | NS | NS |
| • Other | 4 (6) | 7 (9) | NS | 1 (4) | NS | NS |
| • Unknown | 56 (84) | 54 (68) | 0.02 | 22 (81) | NS | NS |
| Known preoperative nephrology referral (%) | 17 (25) | 21 (27) | NS | 20 (74) | <0.001 | <0.001 |

Data are presented as median with interquartile [IQR] or number and frequencies (%).

CKD and AKI stages are defined according to KDIGO 2012 guidelines.

*CKD* chronic kidney disease, *SK* solitary kidney, *RN* radical nephrectomy, *PN* partial nephrectomy, *BMI* body mass index, *ASA* physical status score of American Society of Anesthesiology, *SCr* serum creatinine, *eGFR* estimated glomerular filtration rate.

Preoperative eGFR was assessed a median 12 [5–23] days before surgery, using the CKD-EPI formula in 82% of cases. The median preoperative eGFR in the RN-CKD and PN-CKD groups were 50 [41–55] and 49 [42–55] mL/min/1.73m² respectively, showing no significant difference, and CKD stages were also not significantly different between the two groups. PN-SK patients had more frequently a known preoperative referral to a nephrologist (74%) than patients in the RN-CKD (25%) or PN-CKD (27%) groups (*p* < 0.001 for both; Table 1).

## Surgery and tumor characteristics, surgical and renal complications

Patients in the RN-CKD group had larger tumors than those in the PN-CKD group ($p < 0.001$), with fewer T1 lesions and more T3 lesions ($p < 0.001$ for both), more frequently positive nodal and metastatic status ($p < 0.001$ for both), logically more collecting duct or urothelial carcinomas ($p < 0.001$), and less oncocytomas ($p = 0.04$). The RN-CKD group had more frequently laparoscopic and less frequently robot-assisted procedures than PN-CKD ($p = 0.002$ for both), and the median operative time was longer ($p = 0.03$). PN-SK patients had a longer median operative time compared to PN-CKD patients but had shorter median ischemia time (12 [0–17] vs 18 [11–23] min, $p = 0.009$; Table 2).

Patients in the RN-CKD group had more frequently a severe (Clavien-Dindo >2) intra-operative complication than those in the PN-CKD group ($p = 0.006$), but less frequently a severe postoperative complication ($p = 0.03$), and a longer median length of hospital stay ($p = 0.02$). There was no significant difference in terms of severe intra- or post-operative complications between patients in the PN-SK or PN-CKD groups ($p > 0.05$), but PN-SK patients had a longer median length of hospital stay ($p = 0.002$; Table 2).

There was no significant difference in the frequency of postoperative AKI between those in the RN-CKD group (61%) and in the PN-CKD group (51%), which were mainly KDIGO stage 1 in both, but AKI was more frequent in the PN-SK group (81%) than in the PN-CKD group ($p = 0.0498$), and more severe (KDGIGO stages 2 and 3), requiring more frequently postoperative temporary dialysis (15% vs 1% in the PN-CKD, $p = 0.004$; Table 2). There were 12% of patients who had missing data for the primary outcome or who were lost to follow-up at 1 year.

## Primary outcome 1 year after surgery

After a median follow-up of 12.7 [10.3–15.0] months after surgery, patients in the PN-CKD group presented significantly less frequently the primary composite outcome (CKD progression, or MACE, or death) compared to those in the RN-CKD group (43% vs 71%; $p = 0.007$; Table 3).

## Secondary outcomes 1 year after surgery

One-year CKD progression was significantly less frequent (36% vs 63%; $p = 0.003$), and all-cause mortality at 1 year was lower (1% vs 12%; $p = 0.02$) among patients in the PN-CKD group compared to those in the RN-CKD group. There was no significant difference however in the frequency of MACE or cancer-related death at 1 year (Table 3).

Patients in the RN-CKD group had a lower median eGFR ($p = 0.004$), a greater median decrease in absolute and median relative change of eGFR ($p < 0.001$ for both), and more frequent eGFR <90% of baseline value at 1 year than those in the PN-CKD group ($p < 0.001$). Five patients (8%) in the RN-CKD group and 1 PN-SK patient (4%) started maintenance hemodialysis during this first year, whereas none did after PN-CKD ($p = 0.01$ vs RN-CKD). Rate of nephrologist referral at 1 year was up to 60% after RN-CKD and 61% after PN-CKD groups (Table 3).

## Predictors of the primary and renal outcomes at 1 year

In univariate analysis, variables significantly associated with the main outcome were postoperative AKI, and a larger tumor diameter, whereas sex, age, diabetes mellitus, previous history hypertension, and preoperative eGFR (as a continuous variable) were not. In multivariate analysis, postoperative AKI, and a larger tumor diameter remained independent risk factors for

**Table 2. Surgery and tumor characteristics, and postoperative complications.**

| | CKD patients (*n* = 146) | | *p*-value | SK patients | | *p*-value | |
|---|---|---|---|---|---|---|---|
| | RN-CKD (*n* = 67) | PN-CKD (*n* = 79) | | PN-SK (*n* = 27) | | vs RN-CKD | vs PN-CKD |
| Approach (%): | | | | | | | |
| • Open | 38 (57) | 42 (53) | NS | 20 (74) | | NS | NS |
| • Laparoscopic | 19 (28) | 7 (8) | 0.002 | 1 (4) | | 0.008 | NS |
| • Robot-assisted | 10 (15) | 30 (38) | 0.002 | 6 (22) | | NS | NS |
| Conversion to open surgery (%) | 6 (9) | 1 (1) | NS | 1 (4) | | NS | NS |
| Operative time min | 222 [180–282] | 211 [169–230] | 0.03 | 241 [200–275] | | NS | 0.02 |
| Ischemia type (%): | | | | | | | |
| • Warm ischemia | NA | 69 (87) | NA | 19 (70) | | NA | 0.04 |
| • Cold ischemia | NA | 1 (1) | NA | 1 (4) | | NA | NS |
| • No ischemia | NA | 9 (12) | NA | 7 (26) | | NA | NS |
| Ischemia time min | NA | 18 [11–23] | NA | 12 [0–17] | | 0.009 | NA |
| Surgical margin (%): | | | | | | | |
| • R0 | 57 (85) | 65 (82) | NS | 22 (81) | | NS | NS |
| • R1 or R2 | 7 (10) | 13 (17) | NS | 4 (15) | | NS | NS |
| • Rx | 3 (4) | 1 (1) | NS | 1 (4) | | NS | NS |
| Tumor length mm | 55 [45–80] | 35 [25–55] | <0.001 | 41 [23–50] | | <0.001 | NS |
| Tumor quantity (%): | | | | | | | |
| • 1 | 63 (94) | 71 (90) | NS | 22 (81) | | NS | NS |
| • 2+ | 4 (6) | 8 (10) | NS | 5 (19) | | NS | NS |
| Tumor type (%): | | | | | | | |
| • Clear cell RCC | 37 (55) | 45 (57) | NS | 25 (92) | | <0.001 | <0.001 |
| • Papillary RCC | 6 (9) | 13 (17) | NS | 1 (4) | | NS | NS |
| • Chromo. RCC | 0 | 9 (11) | 0.004 | 1 (4) | | NS | NS |
| • Urothelial | 18 (27) | 0 | <0.001 | 0 | | 0.003 | NS |
| • Oncocytoma | 3 (4) | 12 (15) | 0.04 | 0 | | NS | 0.03 |
| • Other | 2 (3) | 0 | NS | 0 | | NS | NS |
| • No data | 1 (1) | 0 | NS | 0 | | NS | NS |
| Fuhrman/ISUP (%): | | | | | | | |
| • 2 | 6 (9) | 18 (22) | 0.02 | 9 (33) | | 0.003 | NS |
| • 3 | 21 (31) | 32 (41) | NS | 16 (59) | | 0.01 | NS |
| • 4 | 13 (19) | 6 (8) | 0.03 | 1 (4) | | NS | NS |
| • Missing | 27 (40) | 23 (29) | NS | 1 (4) | | <0.001 | 0.007 |
| pTNM staging (%): | | | | | | | |
| • T1 | 11 (16) | 47 (59) | <0.001 | 18 (66) | | <0.001 | NS |
| • T2 | 6 (9) | 3 (4) | NS | 1 (4) | | NS | NS |
| • T3 | 41 (61) | 13 (16) | <0.001 | 5 (19) | | <0.001 | NS |
| • T4 | 2 (3) | 0 | NS | 0 | | NS | NS |
| • Tx | 7 (11) | 16 (21) | NS | 3 (11) | | NS | NS |
| N+ status (%) | 10 (15) | 0 | <0.001 | 1 (4) | | NS | NS |
| M+ status (%) | 11 (16) | 1 (1) | <0.001 | 5 (19) | | NS | <0.001 |
| Adjuvant chemotherapy (%) | 6 (9) | 1 (1) | NS | 1 (4) | | NS | NS |
| Estimated blood loss, mL | 300 [111–725] | 300 [100–755] | NS | 475 [200–981] | | NS | NS |
| Hypotension requiring norepinephrine (%) | 8 (12) | 6 (8) | NS | 3 (11) | | NS | NS |
| Intraoperative transfusion (%) | 23 (34) | 9 (11) | <0.001 | 6 (22) | | NS | NS |
| Severe intraoperative complication (%) | 12 (18) | 3 (4) | 0.006 | 2 (7) | | NS | NS |
| • *Pleural or intestinal wound* | *2 (3)* | *2 (3)* | | *0* | | | |

*(Continued)*

**Table 2.** (Continued)

| | CKD patients (*n* = 146) | | *p*-value | SK patients | *p*-value | |
|---|---|---|---|---|---|---|
| | RN-CKD (*n* = 67) | PN-CKD (*n* = 79) | | PN-SK (*n* = 27) | vs RN-CKD | vs PN-CKD |
| • *Hemorrhage* | *5 (7)* | *1 (1)* | | *1 (4)* | | |
| • *Conversion to open surgery* | *6 (9)* | *1 (1)* | | *1 (4)* | | |
| Postoperative transfusion (%) | 3 (4) | 8 (10) | NS | 4 (15) | NS | NS |
| Severe postoperative complication (%) | 4 (6) | 14 (18) | 0.03 | 8 (30) | 0.002 | NS |
| • *Urinary fistula* | *0* | *2 (3)* | | *3 (11)* | | |
| • *Intestinal obstruction* | *0* | *2 (3)* | | *3 (11)* | | |
| • *Hemorrhage* | *0* | *4 (5)* | | *2 (7)* | | |
| • *Surgical site infection* | *3 (4)* | *8 (10)* | | *2 (7)* | | |
| • *Aspiration pneumonia* | *1 (1)* | *1 (1)* | | *2 (7)* | | |
| Surgical revision (%) | 1 (1) | 9 (11) | 0.02 | 5 (19) | 0.002 | NS |
| Length of hospital stay, days | 6 [4–9] | 5 [3–7] | 0.02 | 8 [4–14] | NS | 0.002 |
| Postoperative AKI (%): | 41 (61) | 40 (51) | NS | 22 (81) | NS | 0.0498 |
| • KDIGO 1 | 34 (51) | 33 (42) | NS | 7 (26) | NS | NS |
| • KDIGO 2 | 1 (1) | 1 (1) | NS | 5 (19) | 0.003 | <0.001 |
| • KDIGO 3 | 6 (9) | 6 (8) | NS | 10 (37) | 0.001 | <0.001 |
| Postoperative acute dialysis (%) | 5 (7) | 1 (1) | NS | 4 (15) | NS | 0.004 |

Data are presented as median with interquartile [IQR] or number and frequencies (%).

pTNM staging is defined according to 7[th] edition UICC 2010.

The severity of intra- and post-operative complications was defined according to Clavien-Dindo classification, with a severe complication defined as Clavien-Dindo > 2.

Detailed causes of the complications are cumulative, i.e. one severe complication of a patient can include more than one etiology.

*CKD* chronic kidney disease, *SK* solitary kidney, *RN* radical nephrectomy, *PN* partial nephrectomy, *RCC* renal cell carcinoma, *Chromo.* chromophobe, *Urothelial* collecting duct or urothelial carcinoma, *AKI* acute kidney injury, *KDIGO* Kidney Disease: Improving Global Outcomes.

**Table 3. Outcomes one year after surgery.**

| | CKD patients | | *p*-value | SK patients | *p*-value | |
|---|---|---|---|---|---|---|
| | RN-CKD | PN-CKD | | PN-SK | vs RN-CKD | vs PN-CKD |
| 1-year primary outcome (%) | 41/58 (71) | 30/70 (43) | 0.002 | 14/24 (58) | NS | NS |
| 1-year CKD progression (%) | 35/56 (63) | 25/70 (36) | 0.003 | 12/23 (52) | NS | NS |
| 1-year MACE (%) | 5/61 (8) | 7/71 (10) | NS | 2/25 (8) | NS | NS |
| 1-year all-cause mortality (%) | 7/61 (12) | 1/71 (1) | 0.02 | 1/25 (4) | NS | NS |
| 1-year cancer-related mortality (%) | 4/60 (7) | 1/71 (1) | NS | 0/25 (0) | NS | NS |
| 1-year eGFR mL/min/1.73m$^2$ | 38 [30–46] | 44 [38–53] | 0.004 | 50 [40–65] | <0.001 | 0.04 |
| 1-year eGFR loss mL/min/1.73m$^2$ | 11 [8–20] | 3 [1–9] | <0.001 | 8 [0–15] | NS | NS |
| 1-year eGFR loss % of baseline | 23 [10–39] | 7 [1–20] | <0.001 | 12 [0–23] | 0.046 | NS |
| 1-year eGFR <90% of baseline (%) | 39/51 (77) | 29/69 (42) | <0.001 | 12/21 (57) | NS | NS |
| 1-year on hemodialysis (%) | 5 (8) | 0 | 0.01 | 1 (4) | NS | NS |
| 1-year known nephrology referral (%) | 34 (60) | 42 (61) | NS | 18 (75) | NS | NS |

Data are presented as median with interquartile [IQR] or number and frequencies (%).

The primary outcome was a composite endpoint including CKD progression, major cardio-vascular events, and all-cause mortality. CKD progression was defined as upgrading of at least 1 stage of CKD (according to KDIGO 2012 guidelines), or initiation of renal replacement therapy, using the closest known eGFR to the 1-year post-operative date. Major cardio-vascular events (MACE) included myocardial infarction, unstable angina, cardiac decompensation, or stroke, reported in the medical record within the first post-operative year

*CKD* chronic kidney disease, *SK* solitary kidney, *RN* radical nephrectomy, *PN* partial nephrectomy, *MACE* major adverse cardio-vascular events, *eGFR* estimated glomerular filtration rate, *KDIGO* Kidney Disease: Improving Global Outcomes.

**Table 4. Logistic regression analysis for the primary outcome.**

|  | Odds ratio | 95% CI | *p*-value |
|---|---|---|---|
| Age, for each additional year | 1.034 | 0.981–1.089 | 0.213 |
| Sex, female (ref) vs male | 0.700 | 0.266–1.842 | 0.470 |
| Diabetes mellitus, no (ref) vs yes | 1.189 | 0.444–3.186 | 0.730 |
| Hypertension, no (ref) vs yes | 1.915 | 0.680–5.394 | 0.219 |
| AKI, no (ref) |  |  | <0.001 |
| • AKI stage KDIGO 1 | 8.681 | 3.230–23.333 | <0.001 |
| • AKI stage KDIGO 2 | 23.495 | 2.334–236.509 | 0.007 |
| • AKI stage KDIGO 3 | 28.874 | 4.769–167.612 | <0.001 |
| Tumor diameter, for each additional cm | 1.214 | 1.017–1.450 | 0.032 |
| Preoperative eGFR, for each additional mL/min |  |  | NS |

Multivariable logistic regression analysis investigating predictors of the primary outcome (CKD progression, MACE, and all-cause mortality) 1 year after surgery.

AKI staging was defined according to KDIGO 2012 guidelines.

*AKI* acute kidney injury, *KDIGO* Kidney Disease: Improving Global Outcomes, *eGFR* estimated glomerular filtration rate.

the main outcome (Table 4). Type of surgery (RN vs PN) could not be included in the model because it showed a too strong colinear association with tumor diameter.

Similarly, logistic regression analysis of independent predictors for CKD progression or absolute eGFR loss at 1 year also identified postoperative AKI and larger tumor diameter (S1 Table).

## Outcomes 1 month after surgery

Analyses of the composite and secondary outcomes 1 month after surgery found similar results, as patients in the PN-CKD group already presented less frequently the composite outcome compared to those in the RN-CKD group ($p = 0.03$), had a higher median eGFR value ($p = 0.01$) and lower median eGFR loss ($p < 0.001$; S2 Table and S1 Fig).

## Discussion

In this retrospective cohort study of patients with moderate-to-severe CKD, we report that PN compared to RN led to less frequent adverse outcomes, defined as a composite endpoint including progression of CKD, MACE or death, 1 year after surgery. This was mostly driven by a less frequent progression of CKD, and lower mortality. In the total population including SK patients, we identified postoperative AKI, and larger tumor size as independent risk factors for the primary outcome, CKD progression, and eGFR loss 1 year after surgery, whereas preoperative eGFR was not. However, the influence of type of surgery was not included in the multivariate analysis owing to a too strong collinearity with tumor size.

The greater benefit of PN compared to RN reported herein is consistent with the accumulated data in patients with normal renal function [1–3]. However, the literature is scarce in CKD patients; some studies have reported a greater benefit of PN on CKD progression for patients with stage 3A CKD but not for stage 3B [23] or with stage 4 CKD [24], and no significant difference in mortality [23, 24]. Conversely, a study found no significant difference between RN and PN in the rate of eGFR decrease for patients with moderate CKD, and no significant difference in overall mortality [6], but the major limitation of this study was the frequency of missing data that was up to two-thirds at the end of follow-up.

The strength of the study presented herein is that it included patients treated after the implementation of recent surgical approaches, with few patients with missing data or lost to follow-up. The present study thus reflects recent surgical approaches, which may notably explain the shorter ischemic time that we report compared to previous studies on PN [24]. Moreover, in all of these previously published cohorts the study period was over 20 years (from the 1980s to 2008 [23], 2014 [6] or 2015 [24]) during which surgical procedures were either open only [24] or open and laparoscopic [23], or were unreported [6], and ischemia time during PN was relatively long (median 40 min) [24] or unreported [6, 23]. Furthermore, the cohort herein is one of the largest comparing PN and RN specifically in patients with moderate-to-severe pre-existing CKD, alongside patients with SK. It is noteworthy that we included SK patients regardless of their CKD staging, including a few patients with stage 1 and 2, i.e. even with eGFR $> 60$ mL/min/1.73m$^2$. This was motivated by the fact that SK condition in itself is considered a CKD, even when impairment of eGFR has not occurred yet [30]. Having a SK, even with normal renal function, has indeed been proved as an independent risk factor of renal function deterioration leading to progression of CKD [31]. Likewise, American Urological Association (AUA) and European Association of Urology (EAU) guidelines also associate SK patients regardless of renal function along with patients with renal dysfunction (moderate to severe CKD), as they classify them similarly as having an "imperative" indication of nephron sparing surgery [2, 3]. In the present study, the high incidence of AKI and of the primary outcome in SK patients emphasizes their great susceptibility to renal lesions during and after partial nephrectomy, in the short and long term. Precipitating SK patients from CKD stages with normal to impaired functions is a particularly concerning event, as this will ultimately exposes them to potential complications due to CKD.

AKI during hospital stay is a known strong independent predictor of incident CKD or CKD progression [32], and we report it here as a major predictor of progression of CKD, MACE or death (the primary outcome). Previous studies also identified AKI and tumor size as predictors of worse renal function after PN [15, 17, 33, 34] or after RN [35], but in patients with mainly normal renal function. None of the published studies comparing PN and RN in patients with moderate or severe CKD assessed postoperative AKI [6, 23, 24]. We report herein a frequency of AKI higher than that reported in previous studies (from 25 to 40%) in participants with predominantly normal baseline renal function [15–17]. The frequency of AKI was even higher herein in SK patients, and more severe, whereas it was reported to range from 15 to 30% in previous studies with a comparable population [34, 36, 37]. These studies probably underdiagnosed AKI due to restrictive and less sensitive non-standardized definitions, while in our study both baseline eGFR and AKI were appropriately assessed: creatinine assays were mostly based on enzymatic techniques, and eGFR was estimated using the most recent and accurate CKD-EPI formula [38]. Furthermore, we defined AKI according to current KDIGO criteria that better identify subgroups at higher risk of complications. This is also emphasized by the fact that most of the AKI we detected were mild KDIGO stage 1 but still independently predicted worse outcomes, confirming a more sensitive diagnosis that yet remains clinically significant.

We found no association between preoperative eGFR and the primary outcome, CKD progression, or eGFR loss. This was, however, reported in previous studies as a risk factor for renal degradation in patients mainly with normal renal function [33, 35, 39] and in one study with only severe CKD patients [24], but as a categorical variable whereas herein it was considered as a continuous variable which might have influenced the results. More importantly, postoperative AKI was not evaluated and not included in the multivariate analysis, contrary to herein. However, the previously identified risk factors for postoperative AKI after PN (preoperative eGFR [15], tumor size [16, 29], ischemia time [16, 17, 29], operative time [15],

comorbidities [age, sex, body mass index, diabetes mellitus, hypertension] [15–17]), most of which were included as covariates in the multivariate logistic and linear regressions. Further studies should therefore investigate predictors of AKI specifically in CKD patients, to identify possible means of nephroprotection for them.

The duration of follow-up is heterogeneous between previous studies making comparison difficult. We chose a 1-year endpoint since it has been proposed that eGFR should be evaluated at least 1 month and 1 year after surgery [40]. Furthermore, it has been previously reported that no significant degradation of renal function occurs from a median of 47 days [15] to 5 months [41] after PN, until a follow-up of 4 years. This suggests that CKD progression occurs predominantly during the first postoperative year, and the importance of early evaluation is supported by the difference in the composite and renal outcomes in the secondary analysis as early as 1 month after PN or RN.

The present study has limitations. Due to its retrospective nature, we cannot rule out selection bias, and bias in data collection. Its monocentric setting limits the generalizability of the results in other different population of CKD patients undergoing nephrectomy. Even though we conducted multivariate analysis, we could not take into account other unknown confounders, or known but unavailable, such as the presence of proteinuria on the risk of CKD progression [42] and use of ACEi/ARBs, or post-operative use of contrast or other nephrotoxic agents. Furthermore, ischemia time could not be analyzed as a potential predictor of the primary or renal outcomes because it was only applicable in PN, and not in RN. In addition, we included every renal tumor requiring surgery, some potentially associated with worse prognosis in the RN group, notably with a higher TNM staging.

In conclusion, we report that in patients with moderate to severe CKD, postoperative AKI was a major independent predictor of the composite outcome including CKD progression, MACE, or death, one year after surgery. Further studies are needed to identify early determinants of AKI in this population. PN led to better outcomes compared to RN, but these results need to be confirmed as the type of surgery could not be integrated in multivariate analysis.

## Supporting information

**S1 Fig. Evolution of eGFR from pre-operative to 1-month and 1-year post-operative values.** Data are presented as median with interquartiles. * $p < 0.05$; *** $p < 0.001$ (paired t-tests). *eGFR* estimated glomerular filtration rate, *RN* radical nephrectomy, *PN* partial nephrectomy, *SK* solitary kidney.
(TIF)

**S2 Fig. Evolution of eGFR from pre-operative to 1 month and 1 year post-operative values, in patients who experienced post-operative acute kidney injury regardless of its stage (All AKI), KDIGO stage 1 (AKI 1), KDIGO stage 2 (AKI 2), KDIGO stage 3 (AKI 3), or in those who did not (No AKI).** Data are presented as median with interquartiles. * $p < 0.05$; *** $p < 0.001$ (paired t-tests). *eGFR* estimated glomerular filtration rate, *AKI* acute kidney injury, *KDIGO* Kidney Disease: Improving Global Outcomes.
(TIF)

**S1 Table. Linear and logistic regression analysis for secondary outcomes.** Multivariable logistic regression and linear regression analysis investigating predictors of respectively CKD progression or of absolute the eGFR loss 1 year after surgery. AKI staging was defined according to KDIGO 2012 guidelines. *CKD* chronic kidney disease, *AKI* acute kidney injury, *KDIGO* Kidney Disease: Improving Global Outcomes, *eGFR* estimated glomerular filtration rate.
(PDF)

**S2 Table. One-month postoperative renal outcomes.** Data are presented as median with interquartile [IQR] or number and frequencies (%). *CKD* chronic kidney disease, *SK* solitary kidney, *RN* radical nephrectomy, *PN* partial nephrectomy, *MACE* major adverse cardio-vascular outcome, *eGFR* estimated glomerular filtration rate.
(PDF)

**S1 Dataset. Minimal anonymized data set.**
(XLSX)

## Acknowledgments

We thank Dr Philip Robinson (Direction de la Recherche Clinique et de l'Innovation, Hospices Civils de Lyon) for his help in manuscript preparation and proofreading. We thank Laura Ratenet for her technical support in the process of data collection.

## Author Contributions

**Conceptualization:** Maxime Schleef, Marc Colombel, Lionel Badet, Fitsum Guebre-Egziabher.

**Data curation:** Maxime Schleef.

**Formal analysis:** Maxime Schleef, Pascal Roy.

**Funding acquisition:** Sandrine Lemoine.

**Investigation:** Maxime Schleef.

**Methodology:** Maxime Schleef, Pascal Roy, Marc Colombel, Fitsum Guebre-Egziabher.

**Project administration:** Fitsum Guebre-Egziabher.

**Resources:** Sandrine Lemoine, Philippe Paparel, Marc Colombel, Lionel Badet.

**Software:** Pascal Roy, Marc Colombel.

**Supervision:** Marc Colombel, Fitsum Guebre-Egziabher.

**Validation:** Fitsum Guebre-Egziabher.

**Visualization:** Maxime Schleef.

**Writing – original draft:** Maxime Schleef.

**Writing – review & editing:** Pascal Roy, Sandrine Lemoine, Lionel Badet, Fitsum Guebre-Egziabher.

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
