## [Decision Letter · Decision Letter 0]

17 Oct 2023

PONE-D-23-29280Renal and major clinical outcomes and their determinants after nephrectomy in patients with pre-existing chronic kidney disease: a retrospective cohort studyPLOS ONE

Dear Dr. Schleef,

Thank you for submitting your manuscript to PLOS ONE. After careful consideration, we feel that it has merit but does not fully meet PLOS ONE’s publication criteria as it currently stands. Therefore, we invite you to submit a revised version of the manuscript that addresses the points raised during the review process.

We look forward to receiving your revised manuscript.

Kind regards,

Yudai Ishiyama

Academic Editor

PLOS ONE

Additional Editor Comments:

The editor have several additional comments to the reviewers' notes.

First, the authors’ decision to include single-kidney patients regardless of their CKD status seems illogical and makes their message vague and unsolid. They should either exclude this population or provide a clearer reason for its inclusion.

Second, if surgical method (radical vs partial) could not be included in the multivariable model due to a strong correlation with tumor size, the statement ‘PN was associated with lower risk of CKD progression, MACE, or death, one year after surgery, compared to RN’ is an overstatement, especially in the abstract.

Reviewers' comments:

Reviewer's Responses to Questions

**Comments to the Author**

1. Is the manuscript technically sound, and do the data support the conclusions?

Reviewer #1: Yes

Reviewer #2: Yes

2. Has the statistical analysis been performed appropriately and rigorously? 

Reviewer #1: I Don't Know

Reviewer #2: Yes

3. Have the authors made all data underlying the findings in their manuscript fully available?

Reviewer #1: Yes

Reviewer #2: Yes

4. Is the manuscript presented in an intelligible fashion and written in standard English?

Reviewer #1: Yes

Reviewer #2: Yes

5. Review Comments to the Author

Reviewer #1: I have some questions for authors.

First, please show us the criteria for choosing PN or RN in author's institution. I want to know author's entire clinical records of preoperative clinical T stage, median tumor size for culculating using CT scan.

Second, I do not understand for including SK patients with CKD1-2 stage. If authors analyze the comparinson of renal outcomes between PN and RN in CKD≥3 patients, you do not have to compare for three groups.

Finally, it would be better to include postoperative taking ACEi/ARBs or nephtotoxic agents as a factor in the multivariable analysis.

Reviewer #2: The study appears to be well-conducted with a clear methodology and relevant outcomes. The results provide valuable insights into the effects of renal surgeries on CKD patients. However, as with all retrospective studies, there's potential for selection bias and reliance on the accuracy of previously recorded data. It would be beneficial to see a prospective study or randomized controlled trial in the future to validate these findings further. Additionally, the generalizability of the results might be limited to the specific population and setting of the study. Overall, the study adds valuable knowledge to the field and has potential implications for clinical practice

6. PLOS authors have the option to publish the peer review history of their article (what does this mean?). If published, this will include your full peer review and any attached files.

Reviewer #1: No

Reviewer #2: No

---

## [Author Response · Author response to Decision Letter 0]

26 Nov 2023

Additional Editor Comments, Comment 1: 

First, the authors’ decision to include single-kidney patients regardless of their CKD status seems illogical and makes their message vague and unsolid. They should either exclude this population or provide a clearer reason for its inclusion.

Authors’ reply: 

Our decision to include solitary kidney (SK) patients regardless of their chronic kidney disease (CKD) status may seem counter-intuitive at first, but we believe it can provide valuable information and that this population should not be excluded. SK patients were often excluded in previous studies focusing on outcomes after partial nephrectomy, and therefore analyzed in their own separate studies, and data about these patients are scarce. In the present study, we wanted to focus specifically on CKD patients to test it their outcomes after nephrectomy were consistent with what has been reported in general population, with patients mostly with a normal renal function, and usually with the exclusion of severe CKD. 

Foremost, SK condition in itself is, and should be, considered a CKD, and recognized as such even when impairment of glomerular filtration rate (GFR) has not occurred yet. In Kidney Disease: Improving Global Outcomes (KDIGO) guidelines (Kidney International Supplements (2013) 3, 5–14), CKD is defined as chronic abnormalities of kidney function (GFR < 60 mL/min/1.73m²) OR kidney structure as well, including structural abnormalities detected by imaging such as anatomical or functional solitary kidney. Even when GFR remains normal, SK should therefore be classified as CKD stage 1-2, reflecting the increased susceptibility of renal complications such as acute kidney injury, GFR deterioration, and CKD progression, and in order to emphasize the importance of nephroprotection. Having a SK, even with normal renal function, has indeed been proved as an independent risk factor of renal function deterioration leading to progression of CKD (Kim et al., 2019, Eur J Epidemiol, doi: 10.1007/s10654-019-00520-7).

Likewise, American Urological Association (AUA) and European Association of Urology (EAU) guidelines (respectively, doi:10.1016/j.juro.2017.04.100 and doi:10.1016/j.eururo.2019.02.011) also associate SK patients regardless of renal function along with patients with renal dysfunction (moderate to severe CKD), as they classify them similarly as having an “imperative” indication of nephron sparing surgery (as opposed to an “elective” indication) to preserve their renal parenchyma and GFR. For example, in the RECORd1 cohort, patients presenting with an imperative indication of PN, including moderate to severe CKD and SK regardless of renal function, were analyzed together in the same subgroup (Minervini et al. 2019, Minerva Urologica E Nefrologica, doi:10.23736/S0393-2249.18.03202-2). In this cohort, imperative surgery (that is patients with CKD and SK regardless of renal function) was a significant predictive risk factor of intra-operative complications, followed by higher risk of overall postoperative complications. 

Furthermore, among the 11 SK patients with CKD stage 1-2 in our cohort, it is noteworthy that 8 (72%) of them experienced post-operative AKI. Consequently, 7 (63%) of them also presented the primary outcome at 1 year, with a progression of CKD, which is a serious and concerning event in this population. This emphasizes their great susceptibility to renal lesions during and after partial nephrectomy, in the short and long term, precipitating them from CKD stages with normal to impaired functions, and ultimately exposing them to potential complications due to CKD.

For all these reasons, we are convinced that all SK patients, even with CKD stage 1-2, are a valuable addition to this paper and its conclusion, and their inclusion provide useful data for practitioners managing renal tumor. The importance of postoperative acute kidney injury as a predictive risk factor of poor renal outcomes should be particularly concerning in daily practice, especially with SK patients as showed by its high incidence in our cohort, and a normal renal function in SK patients should not be falsely reassuring in their preoperative outcome. We believe that the exclusion of SK patients with CKD stage 1-2 would convey a wrong message.

As requested, we detailed more these reasons in the discussion section of the manuscript as follow: “Furthermore, the cohort herein is one of the largest comparing PN and RN specifically in patients with moderate-to-severe pre-existing CKD, alongside patients with SK. It is noteworthy that we included SK patients regardless of their CKD staging, including a few patients with stage 1 and 2, i.e. even with eGFR > 60 mL/min/1.73m². This was motivated by the fact that SK condition in itself is considered a CKD, even when impairment of eGFR has not occurred yet (30). Having a SK, even with normal renal function, has indeed been proved as an independent risk factor of renal function deterioration leading to progression of CKD (31). Likewise, American Urological Association (AUA) and European Association of Urology (EAU) guidelines also associate SK patients regardless of renal function along with patients with renal dysfunction (moderate to severe CKD), as they classify them similarly as having an “imperative” indication of nephron sparing surgery (2,3). In the present study, the high incidence of AKI and of the primary outcome in SK patients emphasizes their great susceptibility to renal lesions during and after partial nephrectomy, in the short and long term. Precipitating SK patients from CKD stages with normal to impaired functions is a particularly concerning event, as this will ultimately exposes them to potential complications due to CKD.”.

Additional Editor Comments, Comment 2: 

Second, if surgical method (radical vs partial) could not be included in the multivariable model due to a strong correlation with tumor size, the statement ‘PN was associated with lower risk of CKD progression, MACE, or death, one year after surgery, compared to RN’ is an overstatement, especially in the abstract.

Authors’ reply: 

Surgical technique could indeed not be included in the multivariate model due to collinearity with tumor size, as it would have been statistically inappropriate. We understand the editor’s concern about this statement, so we modified the abstract and deleted the following statement “In moderate-to-severe CKD patients, PN was associated with lower risk of CKD progression, MACE, or death, one year after surgery, compared to RN. Postoperative AKI after PN or RN was the major independent determinant of worse outcomes (CKD progression, MACE, or death) one year after surgery.” and the conclusion of the manuscript was modified as follow: “In conclusion, we report that in patients with moderate to severe CKD, postoperative AKI was a major independent predictor of the composite outcome including CKD progression, MACE, or death, one year after surgery. Further studies are needed to identify early determinants of AKI in this population. PN led to better outcomes compared to RN, but these results need to be confirmed as the type of surgery could not be integrated in multivariate analysis.”.

 

Reviewer 1, Comment 1: 

First, please show us the criteria for choosing PN or RN in author's institution. I want to know author's entire clinical records of preoperative clinical T stage, median tumor size for calculating using CT scan.

Authors’ reply: 

The criteria for choosing PN or RN was based on french guidelines at the time (doi: 10.1016/S1166-7087(13)70055-1 and doi: 10.1016/S1166-7087(16)30702-3), in which PN was the first option whenever feasible, similar to and still in accordance with current french, european and american guidelines (respectively, doi: 10.1016/S1166-7087(20)30749-1, doi: 10.1016/j.eururo.2019.02.011, and doi: 10.1016/j.juro.2017.04.100). We therefore added in the methods section of the manuscript the following statement: “Inclusion criteria were any surgical procedure (RN or PN) for a renal tumor from January 2013 to December 2018, with a baseline estimated glomerular filtration rate (eGFR) < 60 mL/min/1.73m² prior to surgery (RN-CKD and PN-CKD groups), or a PN for a renal tumor on a SK regardless of eGFR (PN-SK group). The criteria for choosing PN or RN were based on french guidelines at the time, in which PN was the first option whenever feasible (26).”.

Tumor size showed in Table 2 in the manuscript was measured on pathological exam report after the tumor was removed. Tumor size according to CT scan evaluation was also available, whenever it was reported in surgeons’ preoperative consultations (however, we did not have direct acces to CT scan images, CT scan reports results), so we evaluated pre operative clinical T stage based on these informations with the following results:

 CKD patients (n = 146) p-value SK patients p-value

 RN-CKD (n = 67) PN-CKD (n = 79) PN-SK (n = 27) vs RN-CKD vs PN-CKD

Tumor length on preoperative imaging, mm 70 (44-99) 36 (26-52) <0.001 35 (22-50) <0.001 NS

cTNM staging (%): 

- T1 18 (27) 68 (86) <0.001 23 (85) <0.001 NS

- T2 13 (19) 7 (9) NS 1 (4) NS NS

- T3 22 (33) 0 <0.001 1 (4) 0.003 NS

- T4 0 0 NA 0 NA NA

- Tx 14 (21) 4 (5) 0.004 2 (7) NS NS

These results are very similar to those we presented in Table 2. We believe that Table 2 already contains a lot of informations, but if the Editor would like us to do so, we suggest to add it to Table 2.

Reviewer 1, Comment 2: 

Second, I do not understand for including SK patients with CKD1-2 stage. If authors analyze the comparison of renal outcomes between PN and RN in CKD≥3 patients, you do not have to compare for three groups.

Authors’ reply: 

Our decision to include solitary kidney (SK) patients regardless of their chronic kidney disease (CKD) status may seem counter-intuitive at first, but we believe it can provide valuable information and that this population should not be excluded. SK patients were often excluded in previous studies focusing on outcomes after partial nephrectomy, and therefore analyzed in their own separate studies, and data about these patients are scarce. In the present study, we wanted to focus specifically on CKD patients to test it their outcomes after nephrectomy were consistent with what has been reported in general population, with patients mostly with a normal renal function, and usually with the exclusion of severe CKD. 

Foremost, SK condition in itself is, and should be, considered a CKD, and recognized as such even when impairment of glomerular filtration rate (GFR) has not occurred yet. In Kidney Disease: Improving Global Outcomes (KDIGO) guidelines (Kidney International Supplements (2013) 3, 5–14), CKD is defined as chronic abnormalities of kidney function (GFR < 60 mL/min/1.73m²) OR kidney structure as well, including structural abnormalities detected by imaging such as anatomical or functional solitary kidney. Even when GFR remains normal, SK should therefore be classified as CKD stage 1-2, reflecting the increased susceptibility of renal complications such as acute kidney injury, GFR deterioration, and CKD progression, and in order to emphasize the importance of nephroprotection. Having a SK, even with normal renal function, has indeed been proved as an independent risk factor of renal function deterioration leading to progression of CKD (Kim et al., 2019, Eur J Epidemiol, doi: 10.1007/s10654-019-00520-7).

Likewise, American Urological Association (AUA) and European Association of Urology (EAU) guidelines (respectively, doi:10.1016/j.juro.2017.04.100 and doi:10.1016/j.eururo.2019.02.011) also associate SK patients regardless of renal function along with patients with renal dysfunction (moderate to severe CKD), as they classify them similarly as having an “imperative” indication of nephron sparing surgery (as opposed to an “elective” indication) to preserve their renal parenchyma and GFR. For example, in the RECORd1 cohort, patients presenting with an imperative indication of PN, including moderate to severe CKD and SK regardless of renal function, were analyzed together in the same subgroup (Minervini et al. 2019, Minerva Urologica E Nefrologica, doi:10.23736/S0393-2249.18.03202-2). In this cohort, imperative surgery (that is patients with CKD and SK regardless of renal function) was a significant predictive risk factor of intra-operative complications, followed by higher risk of overall postoperative complications. 

Furthermore, among the 11 SK patients with CKD stage 1-2 in our cohort, it is noteworthy that 8 (72%) of them experienced post-operative AKI. Consequently, 7 (63%) of them also presented the primary outcome at 1 year, with a progression of CKD, which is a serious and concerning event in this population. This emphasizes their great susceptibility to renal lesions during and after partial nephrectomy, in the short and long term, precipitating them from CKD stages with normal to impaired functions, and ultimately exposing them to potential complications due to CKD.

For all these reasons, we are convinced that all SK patients, even with CKD stage 1-2, are a valuable addition to this paper and its conclusion, and their inclusion provide useful data for practitioners managing renal tumor. The importance of postoperative acute kidney injury as a predictive risk factor of poor renal outcomes should be particularly concerning in daily practice, especially with SK patients as showed by its high incidence in our cohort, and a normal renal function in SK patients should not be falsely reassuring in their preoperative outcome. We believe that the exclusion of SK patients with CKD stage 1-2 would convey a wrong message.

As requested, we detailed more these reasons in the discussion section of the manuscript as follow: “Furthermore, the cohort herein is one of the largest comparing PN and RN specifically in patients with moderate-to-severe pre-existing CKD, alongside patients with SK. It is noteworthy that we included SK patients regardless of their CKD staging, including a few patients with stage 1 and 2, i.e. even with eGFR > 60 mL/min/1.73m². This was motivated by the fact that SK condition in itself is considered a CKD, even when impairment of eGFR has not occurred yet (30). Having a SK, even with normal renal function, has indeed been proved as an independent risk factor of renal function deterioration leading to progression of CKD (31). Likewise, American Urological Association (AUA) and European Association of Urology (EAU) guidelines also associate SK patients regardless of renal function along with patients with renal dysfunction (moderate to severe CKD), as they classify them similarly as having an “imperative” indication of nephron sparing surgery (2,3). In the present study, the high incidence of AKI and of the primary outcome in SK patients emphasizes their great susceptibility to renal lesions during and after partial nephrectomy, in the short and long term. Precipitating SK patients from CKD stages with normal to impaired functions is a particularly concerning event, as this will ultimately exposes them to potential complications due to CKD.”.

Reviewer 1, Comment 3: 

Finally, it would be better to include postoperative taking ACEi/ARBs or nephtotoxic agents as a factor in the multivariable analysis.

Authors’ reply: 

We were indeed limited by the number of variables that could reasonably be included in the multivariate analysis, given the limitied size of our cohort (still one of the main contemporary cohort on this specific setting of CKD) and to stay statistically sound. We tried to integrate in priority variables that had previously been described as independent risk factors of poor outcomes. Nephroprotective medications such as ACEi and ARBs, and the control of proteinuria, are also known cornerstones of long term renal function preservation in CKD, among others (nephrotoxic agents eviction, cardiovascular risk factors prevention…) and could have been valuable covariates as reviewer 1 rightfully proposed it. Unfortunately, these data could not be obtained in our retrospective cohort. Only anesthesiologic, surgical and hospitalization records (demographic, past medical history, tumor and surgery characteristics, complications during hospitalization…) were acce

---

## [Editor Report · Decision Letter 1]

8 Jan 2024

PONE-D-23-29280R1Renal and major clinical outcomes and their determinants after nephrectomy in patients with pre-existing chronic kidney disease: a retrospective cohort studyPLOS ONE

Dear Dr. Schleef,

Thank you for submitting your manuscript to PLOS ONE. After careful consideration, we feel that it has merit but does not fully meet PLOS ONE’s publication criteria as it currently stands. Therefore, we invite you to submit a revised version of the manuscript that addresses the points raised during the review process.

We look forward to receiving your revised manuscript.

Kind regards,

Yudai Ishiyama

Academic Editor

PLOS ONE

Additional Editor Comments:

The authors have adequately addressed the previous comments, and the manuscript has been revised accordingly.

There are some additional inquiries necessary before considering this manuscript for publication in PLOS One.

#1 The initial review did not mention this, but it seems a significant portion of patients in the RN-CKD cohort had a final diagnosis of urothelial carcinoma. It's important to confirm whether the surgical procedure intended from start to finish for these patients was radical nephrectomy, not nephroureterectomy.

#2 It's also worth noting that no patients with urothelial carcinoma were included in the PN-CKD or SK groups. Among those who died of cancer within one year, how many had urothelial carcinoma pathology? If the majority of deceased patients had urothelial carcinoma rather than RCC, comparing these groups might seem inappropriate.

#3 Additionally, how many patients met both criteria for:

A) "1-year cancer-related mortality" and "1-year CKD progression," or

B) "1-year cancer-related mortality" and "1-year MACE"?

If the majority of patients who died of cancer within a year met either criterion A or B, the statistical impact of the current analysis would be substantially affirmed.

#4 Table 2 indicates that a certain proportion of patients received adjuvant systemic therapies. Please provide specific details (regimens) of these therapies, as some drugs may have nephrotoxic effects.

---

## [Author Response · Author response to Decision Letter 1]

21 Feb 2024

Additional Editor Comments, Comment #1: The initial review did not mention this, but it seems a significant portion of patients in the RN-CKD cohort had a final diagnosis of urothelial carcinoma. It's important to confirm whether the surgical procedure intended from start to finish for these patients was radical nephrectomy, not nephroureterectomy.

Authors’ reply: About ¼ of the patients in the RN-CKD group indeed had a final diagnosis of urothelial carcinoma on the pathological exam report. As we explained in the Methods section, we included surgical procedures intended for a renal tumor only (as reported by our surgeons according to the French Hospital Discharge Summaries Database), and of course we did not consider procedures planned for a urinary excretory system or urinary tract tumor. It is however plausible that in these few cases, the tumors were thought to be developing at the expense of the renal parenchyma, close to the pyelic cavities, and later found that those were rather urothelial tumors. The surgeries were still classified in the end “radical nephrectomies” by surgeons, eventually because it was the surgical procedure intended from start to finish. 

Additional Editor Comments, Comment #2: It's also worth noting that no patients with urothelial carcinoma were included in the PN-CKD or SK groups. Among those who died of cancer within one year, how many had urothelial carcinoma pathology? If the majority of deceased patients had urothelial carcinoma rather than RCC, comparing these groups might seem inappropriate.

Authors’ reply: Of the 67 RN-CKD patients, 7 died within one year after surgery. Among them, 3 had a final diagnosis of urothelial carcinoma (about 17% of the 18 patients with urothelial carcinoma): 1 death due to a multi-metastatic evolution (noteworthy the patient had started hemodialysis in-between), 1 death due to a cerebral hemorrhage, 1 death at home due to an unknown cause (it is worth noting that in-between this patient had started hemodialysis, had presented an episode of pulmonary edema and an episode of unstable angina). The 4 other remaining patients had a final diagnosis of renal cell carcinoma (about 11% of the 38 patients with RCC in RN-CKD): 2 deaths due to pulmonary metastasis, 2 deaths due to pulmonary embolism (including one with a vena cava thrombosis).

In comparison, of the 27 PN-SK patients, one died due to aspiration pneumonia secondary to bowel obstruction, and of the 79 PN-CKD patients, one died due to a multi-metastatic evolution; both patients had a final diagnosis of renal cell carcinoma.

We understand the general concern about the comparability of the RN and PN groups, as we chose to include surgeries performed for any type of renal tumor, and not only renal cell carcinomas, and we highlighted it in the discussion section. We chose foremost to investigate the effect of these procedures, regardless of the underlying oncological disease, on renal function, notably through the nephronic reduction, which is minimized by PN although it leads to ischemia-reperfusion injury due to the necessary vascular clamping. As already mentioned in the first round of review, to conclude that PN is better than RN based only on our data could indeed be considered an overstatement. We would rather emphasize the fact that these types of surgeries have deleterious effects on renal function and outcomes, even more in this specific population of CKD patients, but still probably minimized by PN. Another strong message that we try to convey is that postoperative acute kidney injury is highly prevalent in this population and is a strong independent predictor of worse outcomes, and effort should be made to find mean to prevent it. 

Additional Editor Comments, Comment #3: Additionally, how many patients met both criteria for:

A) “1-year cancer-related mortality” and “1-year CKD progression,” or

B) “1-year cancer-related mortality” and “1-year MACE”?

If the majority of patients who died of cancer within a year met either criterion A or B, the statistical impact of the current analysis would be substantially affirmed.

Authors’ reply: Of the entire cohort (173 patients), only 9 patients died, including 5 patients who died of cancer within one year. One had already met CKD progression criteria (and had actually started hemodialysis), none of them had already presented a MACE, with respect to the information available during the retrospective collection of the data.

As stated before, one patient died at home due to an unknown cause but had already started hemodialysis, had presented an episode of pulmonary edema and an episode of unstable angina. The other patients that died from a non-cancer related cause (or were not proven so), had not presented CKD progression or MACE. To note, pulmonary embolism was not defined as MACE in our study.

Eventually, the mortality event was uncommon (9/173 patients (5%)) in the study herein which is underpowered to identify causality or draw any strong conclusion from this event. 

Additional Editor Comments, Comment #4: Table 2 indicates that a certain proportion of patients received adjuvant systemic therapies. Please provide specific details (regimens) of these therapies, as some drugs may have nephrotoxic effects.

Authors’ reply: 8 patients received adjuvant systemic therapies, described thereafter: 

 CKD patients (n = 146) p-value SK patients p-value

 RN-CKD (n = 67) PN-CKD (n = 79) PN-SK (n = 27) vs RN-CKD vs PN-CKD

… 

Adjuvant chemotherapy (%) 6 (9) 1 (1) NS 1 (4) NS NS

- Sunitinib 2 (3) 0 0 

- Pazopanib 1 (1) 0 1 (4) 

- Temsirolimus 1 (1) 0 0 

- Gemcitabine 1 (1) 1 (1) 0 

- Unknown 1 (1) 0 0 

Most of these patients (5) received anti-angiogenic/anti-VEGF therapies, 1 received m-TOR inhibitor, and 2 received Gemcitabine. The exact treatment was unknown in one patient, that was referred to another oncological center closer to his home, and the only reports subsequently found specified that the patient was receiving adjuvant chemotherapy with no precision. All these treatments are known for a rather good tolerance profile on renal function in general, although it is true that they can induce nephrotoxicity in rare cases (thrombotic microangiopathy, hypertension and proteinuria have for example been described). Noteworthy, among these 8 patients, only 3 (38%) met the CKD progression criteria within one year (1 on Temsirolimus, 1 on Sunitinib and 1 on Pazopanib), as opposed to respectively 63%, 36% and 52% of CKD progression in RN-CKD, PN-CKD and PN-SK in general. We found it reasonable to assume that the influence of these adjuvant systemic therapies on renal function, and on the results and conclusion of our study was negligible. If the editor would like us to do so, we can add a phrase to discuss this limitation in the discussion section, such as: “A few patients received adjuvant systemic therapies, which may rise concern about their nephrotoxic effects. However, among them the incidence of CKD progression criteria within one year was not higher than in the rest of the patients, it is therefore reasonable to assume that the influence of these adjuvant systemic therapies on renal function, and on the results and conclusion of our study was negligible”, and/or we can add these details in a supplementary file.

---

## [Editor Report · Decision Letter 2]

27 Feb 2024

Renal and major clinical outcomes and their determinants after nephrectomy in patients with pre-existing chronic kidney disease: a retrospective cohort study

PONE-D-23-29280R2

Dear Dr. Schleef,

We’re pleased to inform you that your manuscript has been judged scientifically suitable for publication and will be formally accepted for publication once it meets all outstanding technical requirements.

Kind regards,

Yudai Ishiyama

Academic Editor

PLOS ONE

Additional Editor Comments (optional):

Authors sufficiently answered all the queries given.

I have no additional comments and therefore recommend accept in the current form.